# Local Aggregative Games

**Vikas K. Garg**
CSAIL, MIT
vgarg@csail.mit.edu

**Tommi Jaakkola**
CSAIL, MIT
tommi@csail.mit.edu

Aggregative games provide a rich abstraction to model strategic multi-agent interactions. We introduce local aggregative games, where the payoff of each player is a function of its own action and the aggregate behavior of its neighbors in a connected digraph. We show the existence of a pure strategy $\epsilon$-Nash equilibrium in such games when the payoff functions are convex or sub-modular. We prove an information theoretic lower bound, in a value oracle model, on approximating the structure of the digraph with non-negative monotone sub-modular cost functions on the edge set cardinality. We also define a new notion of structural stability, and introduce $\gamma$-aggregative games that generalize local aggregative games and admit $\epsilon$-Nash equilibrium that is stable with respect to small changes in some specified graph property. Moreover, we provide algorithms for our models that can meaningfully estimate the game structure and the parameters of the aggregator function from real voting data.

## 1 Introduction

Structured prediction methods have been remarkably successful in learning mappings between input observations and output configurations [1; 2; 3]. The central guiding formulation involves learning a scoring function that recovers the configuration as the highest scoring assignment. In contrast, in a game theoretic setting, myopic strategic interactions among players lead to a Nash equilibrium or locally optimal configuration rather than highest scoring global configuration. Learning games therefore involves, at best, enforcement of local consistency constraints as recently advocated [4].

[4] introduced the notion of contextual potential games, and proposed a dual decomposition algorithm for learning these games from a set of pure strategy Nash equilibria. However, since their setting was restricted to learning undirected tree structured potential games, it cannot handle (a) asymmetries in the strategic interactions, and (b) higher order interactions. Moreover, a wide class of strategic games (e.g. anonymous games [5]) do not admit a potential function and thus locally optimal configurations do not coincide with pure strategy Nash equilibria. In such games, the existence of only (approximate) mixed strategy equilibria is guaranteed [6].

In this work, we focus on learning *local* aggregative games to address some of these issues. In an aggregative game [7; 8; 9], every player gets a payoff that depends only on its own strategy and the aggregate of all the other players' strategies. Aggregative games and their generalizations form a very rich class of strategic games that subsumes Cournot oligopoly, public goods, anonymous, mean field, and cost and surplus sharing games [10; 11; 12; 13]. In a local aggregative game, a player's payoff is a function of its own strategy and the aggregate strategy of its neighbors (i.e. only a subset of other players). We do not assume that the interactions are symmetric or confined to a tree structure, and therefore the game structure could, in general, be a spanning digraph, possibly with cycles.

We consider local aggregative games where each player's payoff is a convex or submodular Lipschitz function of the aggregate of its neighbors. We prove sufficient conditions under which such games admit some pure strategy $\epsilon$-Nash equilibrium. We then prove an information theoretic lower bound that for a specified $\epsilon$, approximating a game structure that minimizes a non-negative monotone submodular cost objective on the cardinality of the edge set may require exponentially many queries under a zero-order or value oracle model. Our result generalizes the approximability of the submodular minimum spanning tree problem to degree constrained spanning digraphs [14]. We argue that this lower bound might be averted with a dataset of multiple $\epsilon$-Nash equilibrium configurations sampled

from the local aggregative game. We also introduce $\gamma$-aggregative games that generalize local aggregative games to accommodate the (relatively weaker) effect of players that are not neighbors. These games are shown to have a desirable stability property that makes their $\epsilon$-Nash equilibria robust to small fluctuations in the aggregator input. We formulate learning these games as optimization problems that can be efficiently solved via branch and bound, outer approximation decomposition, or extended cutting plane methods [17; 18]. The information theoretic hardness results do not apply to our algorithms since they have access to the (sub)gradients as well, unlike the value oracle model where only the function values may be queried. Our experiments strongly corroborate the efficacy of the local aggregative and $\gamma$-aggregative games in estimating the game structure on two real voting datasets, namely, the US Supreme Court Rulings and the Congressional Votes.

## 2  Setting

We consider an $n$-player game where each player $i \in [n] \triangleq \{1, 2, \dots, n\}$ plays a strategy (or action) from a finite set $A_i$. For any strategy profile $a$, $a_i$ denotes the strategy of the $i^{th}$ player, and $a_{-i}$ the strategies of the other players. We are interested in local aggregative games that have the property that the payoff of each player $i$ depends only on its own action and the aggregate action of its neighbors $N_G(i) = \{j \in V(G) : (j, i) \in E(G)\}$ in a connected digraph $G = (V, E)$, where $|V| = n$. Since, the graph is directed, the neighbors need not be symmetric, i.e., $(j, i) \in E$ does not imply $(i, j) \in E$.

For any strategy profile $a$, we will denote the strategy vector of neighbors of player $i$ by $a_{N_G(i)}$. We assume that player $i$ has a payoff function of the form $u_i(a_i, f_G(a, i))$, where $f_G(a, i) \triangleq f(a_{N_G(i)})$ is a local aggregator function, and $u_i$ is convex and Lipschitz in the aggregate $f_G(a, i)$ for all $a_i \in A_i$. Since $f_G(a, i)$ may take only finitely many values, we will assume interpolation between these values such that they form a convex set. We can define the Lipschitz constant of $G$ as

$$\delta(G) \triangleq \max_{i, a_i, a'_{-i}, a''_{-i}} \{u_i(a_i, f_G(a', i)) - u_i(a_i, f_G(a'', i))\}, \tag{1}$$

where the vectors $a'_{-i}$ and $a''_{-i}$ differ in exactly one coordinate. Clearly, the payoff of any player in the network does not change by more than $\delta(G)$ when one of the neighbors changes its strategy. We can now talk about a class of aggregative games characterized by the Lipschitz constant:

$$L(\Delta, n) = \{G : V(G) = n, \delta(G) \leq \Delta\}.$$

A strategy profile $a = (a_i, a_{-i})$ is said to be a *pure strategy $\epsilon$-Nash equilibrium* ($\epsilon$-PSNE) if no player can improve its payoff by more than $\epsilon$ by unilaterally switching its strategy. In other words, any player $i$ cannot gain more than $\epsilon$ by playing an alternative strategy $a'_i$ if the other players continue to play $a_{-i}$. More generally, instead of playing deterministic actions in response to the actions of others, each player can randomize its actions. Then, the distributions over players' actions constitute a *mixed strategy $\epsilon$-Nash equilibrium* if any unilateral deviation could improve the expected payoff by at most $\epsilon$. We will prove the existence of $\epsilon$-PSNE in our setting. We will assume a training set $S = \{a^1, a^2, \dots, a^M\}$, where each $a^i$ is an $\epsilon$-PSNE sampled from our game. Our objective is to recover the game digraph $G$ and the payoff functions $u_i, i \in [n]$ from the set $S$.

The rest of the paper is organized as follows. We first establish some important theoretical paraphernalia on the local aggregative games in Section 3. In Section 4, we introduce $\gamma$-aggregative games and show that $\gamma$-aggregators are structurally stable. We formulate the learning problem in Section 5, and describe our experimental set up and results in Section 6. We state the theoretical results in the main text, and provide the detailed proofs in the Supplementary (Section 7) for improved readability.

## 3  Theoretical foundations

Any finite game is guaranteed to admit a mixed strategy $\epsilon$-equilibrium due to a seminal result by Nash [6]. However, general games may not have any $\epsilon$-PSNE (for small $\epsilon$). We first prove a sufficient condition for the existence of $\epsilon$-PSNE in local aggregative games with small Lipschitz constant. A similar result holds when the payoff functions $u_i(\cdot)$ are non-negative monotone submodular and Lipschitz (see the supplementary material for details).

**Theorem 1.** Any local aggregative game on a connected digraph $G$, where $G \in L(\Delta, n)$ and $\max_i |A_i| \leq m$, admits a $10\Delta\sqrt{\ln(8mn)}$-PSNE.

*Proof.* (Sketch.) The main idea behind the proof is to sample a random strategy profile from a mixed strategy Nash equilibrium of the game, and show that with high probability the sampled profile corresponds to an $\epsilon$-PSNE when the Lipschitz constant is small. The proof is based on a novel application of the Talagrand's concentration inequality. $\square$

Theorem 1 implies the minimum degree $d$ (which depends on number of players $n$, the local aggregator function $A$, Lipschitz constant $\Delta$, and $\epsilon$) of the game structure that ensures the existence of at least one $\epsilon$-PSNE. One example is the following local generalization of binary summarization games [8]. Each player $i$ plays $a_i \in \{0, 1\}$ and has access to an averaging aggregator that computes the fraction of its neighbors playing action 1. Then, the Lipschitz constant of $G$ is $1/k$, where $k$ is the minimum degree the underlying game digraph. Then, an $\epsilon$-PSNE is guaranteed for $k = \Omega(\sqrt{\ln n}/\epsilon)$. In other words, $k$ needs to grow slowly (i.e., only sub-logarithmically) in the number of players $n$.

An important follow-up question is to determine the complexity of recovering the underlying game structure in a local aggregative game with an $\epsilon$-PSNE. We will answer this question in a combinatorial setting with non-negative monotone submodular cost functions on the edge set cardinality. Specifically, we consider the following problem. Given a connected digraph $G(V, E)$, a degree parameter $d$, and a submodular cost function $h : 2^E \to \mathbb{R}^+$ that is normalized (i.e. $h(\emptyset) = 0$) and monotone (i.e. $h(S) \leq h(T)$ for all $S \subseteq T \in 2^E$), we would like to find a spanning directed subgraph[1] $G_s$ of $G$ such that $f(G_s)$ is minimized, the in-degree of each player is at least $d$, and $G_s$ admits some $\epsilon$-Nash equilibrium when players play to maximize their individual payoffs. We first establish a technical lemma that provides tight lower and upper bounds on the probability that a directed random graph is disconnected, and thus extends a similar result for Erdős-Rényi random graphs [25] to the directed setting. The lemma will be invoked while proving a bound for the recovery problem, and might be of independent interest beyond this work.

**Lemma 2.** Consider a directed random graph $DG(n, p)$ where $p \in (0, 1)$ is the probability of choosing any directed edge independently of others. Define $q = 1 - p$. Let $P_n$ be the probability that $DG$ is connected. Then, the probability that $DG$ is disconnected is $1 - P_N = nq^{2(n-1)} + O\left(n^2 q^{3n}\right)$.

We will now prove an information theoretic lower bound for the recovery problem under the *value oracle* model [14]. A problem with an information theoretic lower bound of $\beta$ has the property that any randomized algorithm that approximates the optimum to within a factor $\beta$ with high probability needs to make superpolynomial number of queries under the specified oracle model. In the value oracle model, each query $Q$ corresponds to obtaining the cost/value of any candidate set by issuing $Q$ to the value oracle (which acts as a black-box). We invoke the Yao's minimax principle [28], which states the relation between distributional complexity and randomized complexity. Using Yao's principle, the performance of randomized algorithms can be lower bounded by proving that no deterministic algorithm can perform well on an appropriately defined distribution of hard inputs.

**Theorem 3.** Let $\epsilon > 0$, and $\alpha, \delta \in (0, 1)$. Let $n$ be the number of players in a local aggregative game, where each player $i \in [n]$ is provided with some convex $\Delta$-Lipschitz function $u_i$ and an aggregator $A$. Let $D_n \triangleq D_n(\Delta, \epsilon, A, (u_i)_{i \in [n]})$ be the sufficient in-degree (number of incoming edges) of each player such that the game admits some $\epsilon$-PSNE when the players play to maximize their individual payoffs $u_i$ according to the local information provided by the aggregator $A$. Assume any non-negative monotone submodular cost function on the edge set cardinality. Then for any $d \geq \max\{D_n, n^\alpha \ln n\}/(1 - \alpha)$, any randomized algorithm that approximates the game structure to a factor $n^{1-\alpha}/(1 + \delta)d$ requires exponentially many queries under the value oracle model.

*Proof.* (Sketch.) The main idea is to construct a digraph that has exponentially many spanning directed subgraphs, and define two carefully designed submodular cost functions over the edges of the digraph, one of which is deterministic in query size while the other depends on a distribution. We make it hard for the deterministic algorithm to tell one cost function from the other. This can be accomplished by ensuring two conditions: (a) these cost functions map to the same value on *almost* all the queries, and (b) the discrepancy in the optimum value of the functions (on the optimum query) is massive. The proof invokes Lemma 2, exploits the degree constraint for $\epsilon$-PSNE, argues about the optimal query size, and appeals to the Yao's minimax principle. $\square$

Theorem 3 might sound pessimistic from a practical perspective, however, a closer look reveals why the query complexity turned out to be prohibitive. The proof hinged on the fact that *all* spanning subgraphs with same edge cardinality that satisfied the sufficiency condition for existence of *any* $\epsilon$-PSNE were equally good with respect to our deterministic submodular function, and we created an instance with exponentially such spanning subgraphs. However, we might be able to circumvent Theorem 3 by breaking the symmetry, e.g., by using data that specifies multiple distinct $\epsilon$-Nash equilibria. Then, since the digraph instance would be required to satisfy these equilibria, fooling the deterministic algorithm would be more difficult. Thus data could, in principle, help us avoid the complexity result of Theorem 3. We will formulate optimization problems that would enforce margin separability on the equilibrium profiles, which will further limit the number of potential digraphs and thus facilitate learning the aggregative game. Moreover, the hardness result does not apply to our estimation algorithms that will have access to the (sub)gradients in addition to the function values.

## 4   $\gamma$-Aggregative Games

We now describe a generalization of the local aggregative games, which we call the $\gamma$-aggregative games. The main idea behind these games is that a player $i \in [n]$ may, often, be influenced not only by the aggregate behavior of its neighbors, but also to a lesser extent on the aggregate behavior of the other players, whose influence on the payoff of $i$ decreases with increase in their distance to $i$. Let $d_G(i, j)$ be the number of intermediate nodes on a shortest path from $j$ to $i$ in the underlying digraph $G = (V, E)$. That is, $d_G(i, j) = 0$ if $(j, i) \in E$, and $1 + \min_{k \in V \setminus \{i,j\}} d_G(i, k) + d_G(k, j)$ otherwise. Let $W_G \triangleq \max_{i,j \in V} d_G(i, j)$ be the width of $G$. For any strategy profile $a \in \{0, 1\}^n$ and $t \in \{0, 1, \ldots, W_G\}$, let $I_G^t(i) = \{j : d_G(i, j) = t\}$ be the set of nodes that have exactly $t$ intermediaries on a shortest path to $i$, and let $a_{I_G^t(i)}$ be a strategy profile of the nodes in this set. We define aggregator functions $f_G^t(a, i) \triangleq f(a_{I_G^t(i)})$ that return the aggregate at level $t$ with respect to player $i$. Let $\gamma \in (0, 1)$ be a discount rate. Define the $\gamma$-aggregator function

$$g_G(a, \gamma, \ell, i) \triangleq \sum_{t=0}^{\ell} \gamma^t f_G^t(a, i) / \sum_{t=0}^{\ell} \gamma^t,$$

which discounts the aggregates based on the distance $\ell \in \{0, 1, \ldots, W_G\}$ to $i$. We assume that player $i \in [n]$ has a payoff function of the form $u_i(a_i, \cdot)$, which is convex and $\eta$-Lipschitz in its second argument for each fixed $a_i$. Finally, we define the Lipschitz constant of the $\gamma$-aggregative game as

$$\delta^\gamma(G) \triangleq \max_{i, a_i, a'_{-i}, a''_{-i}} \{u_i(a_i, g_G(a', \gamma, W_G, i)) - u_i(a_i, g_G(a'', \gamma, W_G, i))\},$$

where the vectors $a'_{-i}$ and $a''_{-i}$ differ in exactly one coordinate.

The main criticism of the concept of $\epsilon$-Nash equilibrium concerns lack of stability: if any player deviates (due to $\epsilon$-incentive), then in general, some other player may have a high incentive to deviate as well, resulting in a non-equilibrium profile. Worse, it may take exponentially many steps to reach an $\epsilon$-equilibrium again. Thus, stability of $\epsilon$-equilibrium is an important consideration. We will now introduce an appropriate notion of stability, and prove that $\gamma$-aggregative games admit stable pure strategy $\epsilon$-equilibrium in that any deviation by a player does not affect the equilibrium much.

**Structurally Stable Aggregator (SSA)**: Let $G = (E, V)$ be a connected digraph and $P_G(w)$ be a property of $G$, where $w$ denotes the parameters of $P_G$. Let $\mathcal{A}$ be an aggregator function that depends on $P_G$. Suppose $M = (a_1, a_2, \ldots, a_n)$ be an $\epsilon$-PSNE when $\mathcal{A}$ aggregates information according to $P_G(w)$, where $a_i$ is the strategy of player $i \in V = [n]$. Suppose now $\mathcal{A}$ aggregates information according to $P_G(w')$. Then, $\mathcal{A}$ is a $(\alpha, \beta)_{P,w,w'}$-structurally stable aggregator (SSA) with respect to $G$, where $\alpha$ and $\beta$ are functions of the gap between $w, w'$, if it satisfies these conditions: (a) $M$ is a $(\epsilon + \alpha)$-equilibrium under $P_G(w')$, and (b) the payoff of each player at the equilibrium profile $M$ under $P_G(w')$ is at most $\beta = O(\alpha)$ worse than that under $P_G(w)$.

A SSA with small values of $\alpha$ and $\beta$ with respect to a small change in $w$ is desirable since that would discourage the players from deviating from their $\epsilon$-equilibrium strategy, however, such an aggregator might not exist in general. The following result shows the $\gamma$-aggregator is a SSA.

**Theorem 4.** Let $\gamma \in (0,1)$, and $g_G(\cdot, \cdot, \ell, \cdot)$ be the $\gamma$-aggregator defined above. Let $P_G(\ell)$ be the property "*the number of maximum permissible intermediaries in a shortest path of length $\ell$ in $G$*". Then, $g_G$ is a $(2\eta\kappa_G, \eta\kappa_G)_{P,W_G,L}$- SSA, where $L < W_G$ and $\kappa_G$ depends on $\gamma$ and $W_G - L$.

## 5 Learning formulation

We now formulate an optimization problem to recover the underlying graph structure, the parameters of the aggregator function, and the payoff functions. Let $S = \{a^1, a^2, \ldots, a^M\}$ be our training set, where each strategy profile $a^m \in \{0,1\}^n$ is an $\epsilon$-PSNE, and $a_i^m$ is the action of player $i$ in example $m \in [M]$. Let $f$ be a local aggregator function, and let $a_{N_i}^m$ be the actions of neighbors $N_i$ of player $i \in [n]$ on training example $m$. We will also represent $N$ as a 0-1 adjacency matrix, with the interpretation that $N_{ij} = 1$ implies that $j \in N_i$, and $N_{ij} = 0$ otherwise. We will use the notation $N_{i\cdot} \triangleq \{N_{ij} : j \neq i\}$. Note that since the underlying game structure is represented as a digraph, $N_{ij}$ and $N_{ji}$ need not be equal. Let $h$ be a concave function such that $h(0) = 0$. Then $F_i(h) \triangleq h(|N_i|)$ is submodular since the concave transformation of the cardinality function results in a submodular function. Moreover $F(h) = \sum_{i\in[n]} F_i(h)$ is submodular since it is a sum of submodular functions. We will use $F(h)$ as a sparsity-inducing prior. Several choices of $h$ have been advocated in the literature, including suitably normalized geometric, log, smooth log and square root functions [15].

We would denote the parameters of the aggregator function $f$ by $\theta_f$. The payoff functions will depend on the choice of this parameterization. For a fixed aggregator $f$ (such as the sum aggregator), linear parameterization is one possibility, where the payoff function for player $i \in [n]$ takes the form,

$$u_i^f(a^m, N_{i\cdot}) = a_i^m w_{i1}(w_f f(a_{N_i}^m) + b_f) + (1 - a_i^m)w_{i0}(w_f f(a_{N_i}^m) + b_f),$$

where $w_{i\cdot} = (w_{i0}, w_{i1})^\top$ and $N_{i\cdot}$ denote the independent parameters for player $i$ and $\theta_f = (w_f, b_f)^\top$ are the shared parameters. Our setting is flexible, and we can easily accommodate more complex aggregators instead of the standard aggregators (e.g. sum). Exchangeable functions over sets [16] provide one such example. An interesting instantiation is a neural network comprising one hidden layer, an output sum layer, with tied weights. Specifically, let $W \in \mathbb{R}^{n \times (n-1)}$ where all entries of $W$ are equal to $w_{NN}$. Let $\sigma$ be an element-wise non-linearity (e.g. we used the ReLU function, $\sigma(x) = \max\{x, 0\}$ for our experiments). Then, using the element-wise multiplication operator $\odot$ and a vector $\mathbf{1}$ with all ones, $u_i$ may be expressed as $u_i^{f_{NN}}(a^m, N_{i\cdot}) = a_i^m w_{i1} f_{NN}(a_{N_i}^m) + (1 - a_i^m)w_{i0} f_{NN}(a_{N_i}^m)$, where the permutation invariant neural aggregator, parameterized by $\theta_{f_{NN}} = (w_{NN}, b_{NN})^\top$,

$$f_{NN}(a_{N_i}^m) = \mathbf{1}^\top \sigma(W\, a_{-i}^m \odot N_{i\cdot} + b_{NN}).$$

We could have more complex functions such as deeper neural nets, with parameter sharing, at the expense of increased computation. We believe this versatility makes local aggregative games particularly attractive, and provides a promising avenue for modeling structured strategic settings.

Each $a^m$ is an $\epsilon$-PSNE, so it ensures a locally (near) optimal reward for each player. We will impose a margin constraint on the difference in the payoffs when player $i$ unilaterally deviates from $a_i^m$. Note that $N_i = \{j \in N_{i\cdot} : N_{ij} = 1\}$. Then, introducing slack variables $\xi_i^m$, and hyperparameters $C, C', C_f > 0$, we obtain the following optimization problem in $O(n^2)$ variables:

$$\min_{\theta_f, w_1, \ldots, w_n, N_1, \ldots, N_n, \xi} \quad \frac{1}{2}\sum_{i=1}^n \|w_{i\cdot}\|^2 + \frac{C_f}{2M}\|\theta_f\|^2 + \frac{C'}{n}\sum_{i=1}^n F_i(h) + \frac{C}{M}\sum_{i=1}^n\sum_{m=1}^M \xi_i^m$$

$$\begin{aligned}
s.t. \quad &\forall i \in [n], m \in [M]: &u_i^f(a^m, N_{i\cdot}) - u_i^f(1 - a^m, N_{i\cdot}) \geq e(a^m, a') - \xi_i^m\\
&\forall i \in [n], m \in [M]: &\xi_i^m \geq 0\\
&\forall i \in [n]: &N_{i\cdot} \in \{0,1\}^{n-1},
\end{aligned}$$

where $a^m$ and $a'$ differ in exactly one coordinate, and $e$ is a margin specific loss term, such as Hamming loss $e_H(a, \tilde{a}) = 1\{a \neq \tilde{a}\}$ or scaled 0-1 loss $e_s(a, \tilde{a}) = 1\{a \neq \tilde{a}\}/n$. From a game theoretic perspective, the scaled loss has a natural asymptotic interpretation: as the number of players $n \to \infty$, $e_s(a^m, a') \to 0$, and we get $\forall i \in [n], m \in [M]: u_i^f(a^m, N_{i\cdot}) \geq u_i^f(1 - a^m, N_{i\cdot}) - \xi_i^m$, i.e., each training example $a^m$ is an $\epsilon$-PSNE, where $\epsilon = \max_{i\in[n],m\in[M]} \xi_i^m$.

Once $\theta_f$ are fixed, the problem clearly becomes separable, i.e., each player $i$ can solve an independent sub-problem in $O(n)$ variables. Each sub-problem includes both continuous and binary variables,

and may be solved via branch and bound, outer approximation decomposition, or extended cutting plane methods (see [17; 18] for an overview of these techniques). The individual solutions can be forced to agree on $\theta_f$ via a standard dual decomposition procedure, and methods like alternating direction method of multipliers (ADMM) [19] could be leveraged to facilitate rapid agreement of the continuous parameters $w_f$ and $b_f$. The extension to learning the $\gamma$-aggregative games is immediate.

We now describe some other optimization variants for the local aggregative games. Instead of constraining each player to a hard neighborhood, one might relax the constraints $N_{ij} \in \{0, 1\}$ to $N_{ij} \in [0, 1]$, where $N_{ij}$ might be interpreted as the strength of the edge $(j, i)$. The Lovász convex relaxation of $F$ [20] is a natural prior for inducing sparsity in this case. Specifically, for an ordering of values $|N_{i(0)}| \geq |N_{i(1)}| \ldots \geq |N_{i(n-1)}|$, $i \in [n]$, this prior is given by

$$\Gamma_h(N) = \sum_{i=1}^{n} \Gamma_h(N, i), \text{ where } \Gamma_h(N, i) = \sum_{k=0}^{n-1} [h(k+1) - h(k)]|N_{i(k)}|.$$

Since the transformation $h$ encodes the preference for each degree, $\Gamma_h(N)$ will act as a prior that encourages structured sparsity. One might also enforce other constraints on the structure of the local aggregative game. For instance, an undirected graph could be obtained by adding constraints $N_{ij} = N_{ji}$, for $i \in [n], j \neq i$. Likewise, a minimum in-degree constraint may be enforced on player $i$ by requiring $\sum_j N_{ij} \geq d$. Both these constraints are linear in $N_i$, and thus do not add to the complexity of the problem. Finally, based on cues such as domain knowledge, one may wish to add a degree of freedom by not enforcing sharing of the parameters of the aggregator among the players.

## 6 Experiments

We now present strong empirical evidence to demonstrate the efficacy of local aggregative games in unraveling the aggregative game structure of two real voting datasets, namely, the US Supreme Court Rulings dataset and the Congressional Votes dataset. Our experiments span the different variants for recovering the structure of the aggregative games including settings where (a) parameters of the aggregator are learned along with the payoffs, (b) in-degree of each node is lower bounded, (c) $\gamma$-discounting is used, or (d) parameters of the aggregator are fixed. We will also demonstrate that our method compares favorably with the potential games method for tree structured games [4], even when we relax the digraph setting to let weights $N_{ij} \in [0, 1]$ instead of $\{0, 1\}$ or force the game structure to be undirected by adding the constraints $N_{ij} = N_{ji}$. For our purposes, we used the smoothed square-root concave function, $h(i) = \sqrt{i + 1} - 1 + \alpha i$ parameterized by $\alpha$, the sum and neural aggregators, and the scaled 0-1 loss function $e_s(a, \tilde{a}) = 1\{a \neq \tilde{a}\}/n$. We found our model to perform well across a very wide range of hyperparameters. All the experiments described below used the following setting of values: $\alpha = 1$, $C = 100$, and $C_f = 1$. $C'$ was also set to 0.01 in all settings except when the parameters of the aggregator were fixed, when we set $C' = 0.01\sqrt{n}$.

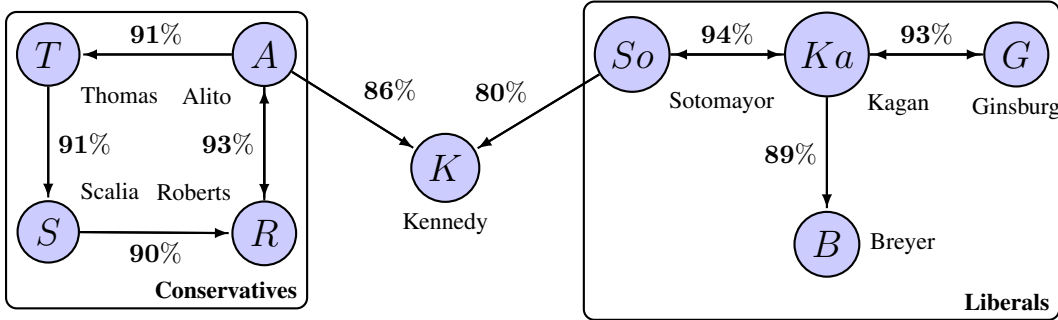

Figure 1: **Supreme Court Rulings (full bench):** The digraph recovered by the local aggregative and $\gamma$-aggregative games ($\ell \leq 2$, all $\gamma$) with the sum aggregator as well as the neural aggregator is consistent with the known behavior of the Justices: conservative and liberal sides of the bench are well segregated from each other, while the moderate Justice Kennedy is positioned near the center. Numbers on the arrows are taken from an independent study [21] on Justices' mutual voting patterns.

## 6.1 Dataset 1: Supreme Court Rulings

We experimented with a dataset containing all non-unanimous rulings by the US Supreme court bench during the year 2013. We denote the Justices of the bench by their last name initials, and add a second character to some names to avoid the conflicts in the initials: Alito (A), Breyer (B), Ginsburg(G), Kennedy (K), Kagan (Ka), Roberts (R), Scalia (S), Sotomayor (So), and Thomas (T). We obtained a binary dataset following the procedure described in [4].

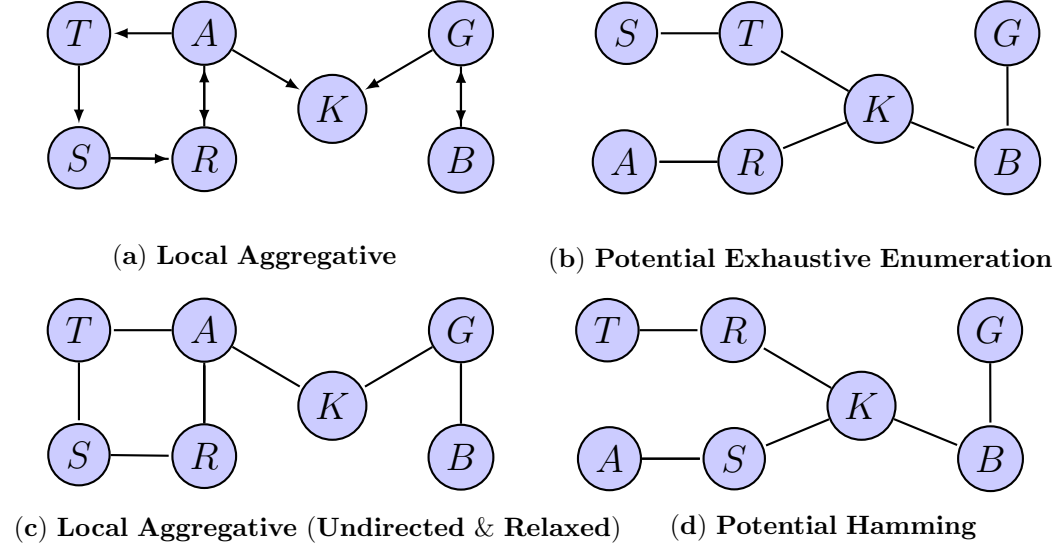

(a) **Local Aggregative**          (b) **Potential Exhaustive Enumeration**

(c) **Local Aggregative (Undirected & Relaxed)**     (d) **Potential Hamming**

Figure 2: **Comparison with the potential games method [4]:** (a) The digraph produced by our method with the sum as well as the neural aggregator is consistent with the expected voting behavior of the Justices on the data used by [4] in their experiments. (c) Relaxing all $N_{ij} \in [0, 1]$ and enforcing $N_{ij} = N_{ji}$ still resulted in a meaningful undirected structure. (b) & (d) The tree structures obtained by the brute force and the Hamming distance restricted methods [4] fail to capture higher order interactions, e.g., the strongly connected component between Justices A, T, S and R.

.

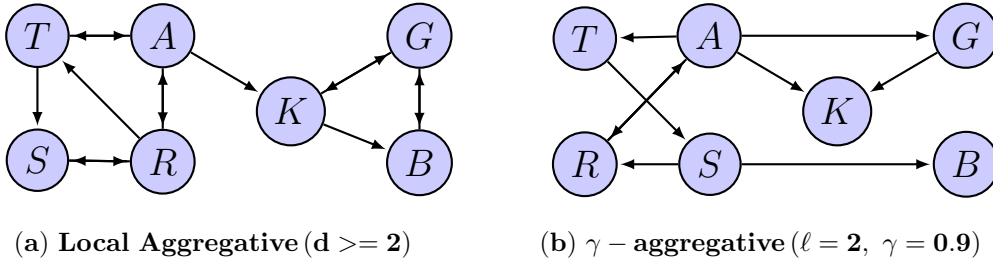

(a) **Local Aggregative** $(d >= 2)$         (b) $\gamma-$ **aggregative** $(\ell = 2, \ \gamma = 0.9)$

Figure 3: **Degree constrained and $\gamma$-aggregative games**: (a) Enforcing the degree of each node to be at least 2 reinforces the intra-republican and the intra-democrat affinity, reaffirming their respective jurisprudences, and (b) $\gamma$-aggregative games also support this observation: the same digraph as Fig. 2(a) is obtained unless $\ell$ and $\gamma$ are set to high values (plot generated with $\ell = 2, \gamma = 0.9$), when the strong effect of one-hop and two-hop neighbors overpowers the direct connection between $B$ and $G$.

Fig. 1 shows the structure recovered by the local aggregative method. The method was able to distinguish the conservative side of the court (Justices A, R, S, and T) from the left side (B, G, Ka, and So). Also, the structure places Justice Kennedy in between the two extremes, which is consistent with his moderate jurisprudence. To put our method in perspective, we also compare the result of applying our method on the same subset of the full bench data that was considered by [4] in their experiments. Fig. 2 demonstrates how the local aggregative approach estimated meaningful structures consistent with the full bench structure, and compared favorably with both the methods of [4]. Finally, Fig. 3(a)

and 3(b) demonstrate the effect of enforcing minimum in-degree constraints in the local aggregative games, and increasing $\ell$ and $\gamma$ in the $\gamma$-aggregative games respectively. As expected, the estimated $\gamma$-aggregative structure is stable unless $\gamma$ and $\ell$ are set to high values when non-local effects kick in. We provide some additional results on the degree-constrained local aggregative games (Fig. 4 ) and the $\gamma$-aggregative games (Fig. 5). In particular, we see that the $\gamma$-aggregative games are indeed robust to small changes in the aggregator input as expected in the light of stability result of Theorem 4.

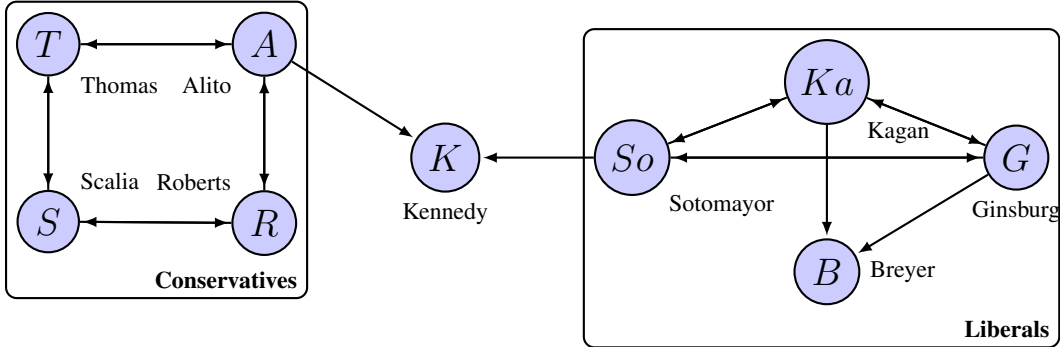

Figure 4: **Degree constrained local aggregative games (full bench):** The digraph recovered by the local aggregative method when the degree of each node was constrained to be at least 2. Clearly, the cohesion among the Justices on the conservative side got strengthened by the degree constraint (likewise for the liberal side of the bench). On the other hand, no additional edges were added between the two sides.

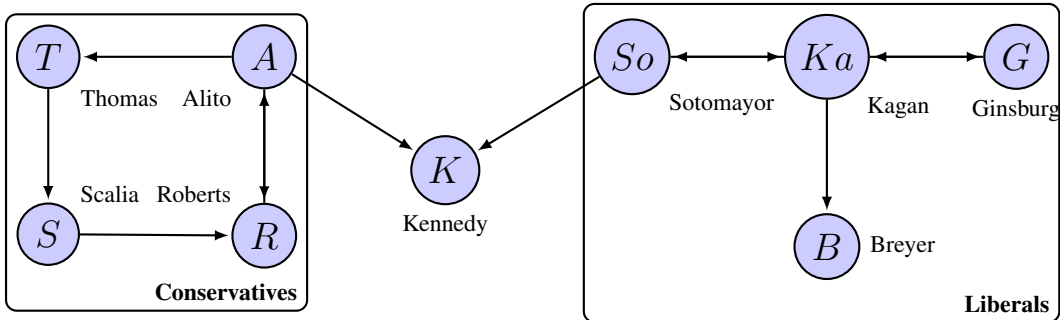

Figure 5: $\gamma$-**Aggregative Games (full bench):** The digraph estimated by the $\gamma$-aggregative method for $\ell = 2, \gamma = 0.9$, and lower values of $\gamma$ and/or $\ell$. Note that an identical structure was obtained by the local aggregative method (Fig. 1). This indicates that despite heavily weighting the effect of the nodes on a shortest path with one or two intermediary hops, the structure in Fig. 1 is very stable. Also, this substantiates our theoretical result about the stability of the $\gamma$-aggregative games.

### 6.2   Dataset 2: Congressional Votes

We also experimented with the Congressional Votes data [22], that contains the votes by the US Senators on all the bills of the 110 US Congress, Session 2. Each of the 100 Senators voted in favor of (treated as 1) or against each bill (treated as 0). Fig. 6 shows that the local aggregative method provides meaningful insights into the voting patterns of the Senators as well. In particular, few connections exist between the nodes in red and those in blue, making the bipartisan structure quite apparent. In some cases, the intra-party connections might be bolstered due to same state affiliations, e.g. Senators Corker (28) and Alexander (2) represent Tennessee. The cross connections may also capture some interesting collaborations or influences, e.g., Senators Allard (3) and Clinton (22) introduced the Autism Act. Likewise, Collins (26) and Carper (19) reintroduced the Fire Grants Reauthorization Act. The potential methods [4] failed to estimate some of these strategic interactions. Likewise, Fig. 7 provides some interesting insights regarding the ideologies of some Senators that follow a more centrist ideology than their respective political affiliations would suggest.

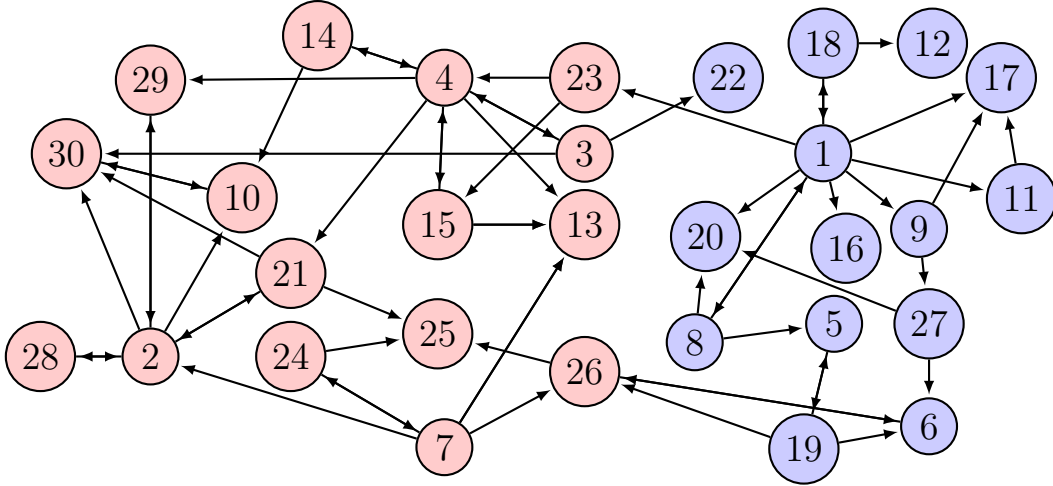

Figure 6: **Comparison with [4] on the Congressional Votes data:** The digraph recovered by local aggregative method, on the data used by [4], when the parameters of the sum aggregator were fixed ($w_f = 1$, $b_f = 0$). The segregation between the Republicans (shown in red) and the Democrats (shown in blue) strongly suggests that they are aligned according to their party policies.

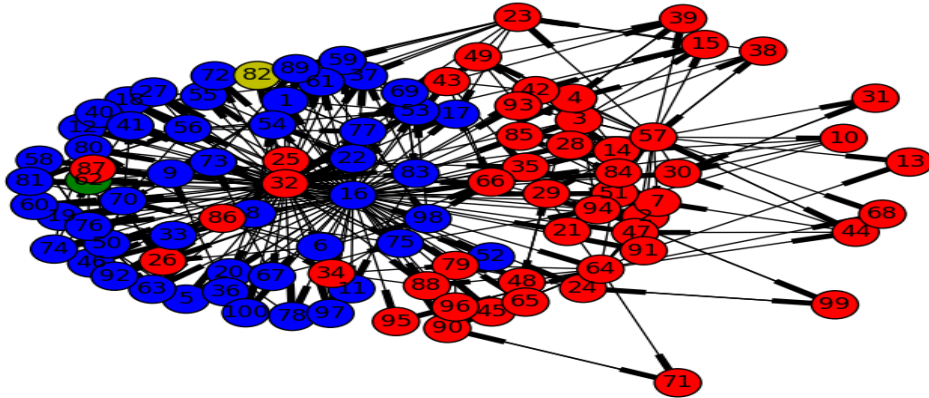

Figure 7: **Complete Congressional Votes data:** The digraph recovered on fixing parameters, relaxing $N_{ij}$ to $[0, 1]$, and thresholding at $0.05$. The estimated structure not only separates majority of the reds from the blues, but also associates closely the then independent Senators Sanders (82) and Lieberman (62) with the Democrats. Moreover, the few reds among the blues generally identify with a more centrist ideology - Collins (26) and Snowe (87) are two prominent examples.

## Conclusion

An overwhelming majority of literature on machine learning is restricted to modeling non-strategic settings. Strategic interactions in several real world systems such as decision/voting often exhibit local structure in terms of how players are guided by or respond to each other. In other words, different agents make rational moves in response to their neighboring agents leading to locally stable configurations such as Nash equilibria. Another challenge with modeling the strategic settings is that they are invariably unsupervised. Consequently, standard learning techniques such as structured prediction that enforce global consistency constraints fall short in such settings (cf. [4]). As substantiated by our experiments, local aggregative games nicely encapsulate various strategic applications, and could be leveraged as a tool to glean important insights from voting data. Furthermore, the stability of approximate equilibria is a primary consideration from a conceptual viewpoint, and the $\gamma$-aggregative games introduced in this work add a fresh perspective by achieving structural stability.

## Footnotes

[1]A spanning directed graph spans all the vertices, and has the property that the (multi)graph obtained by replacing the directed edges with undirected edges is connected.

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
