[Supplementary Material · supp_final.pdf]

# 7 Supplementary (Local Aggregative Games)

We now provide detailed proofs of all the theoretical results stated in the main text.

## 7.1 Existence of $\epsilon$-PSNE in local aggregative games

We will use the following version of Talagrand's inequality that gives a concentration of convex Lipschitz functions around the median.

**Lemma (Talagrand's inequality).** Let $P = \mu_1 \otimes \mu_2 \ldots \otimes \mu_n$ be a product probability measure on the Cartesian product $A = A_1 \times A_2 \ldots \times A_n$ of metric spaces $(A_i, d_i)$ equipped with the $l^1$ metric $d = \sum_{i=1}^{n} d_i$ where each $d_i$ is bounded. Let $a = (a_1, \ldots, a_n)$ be a point in this product space and $F : A \to \mathbb{R}$ be a convex 1-Lipschitz function on $(A, d)$. Then, for any $r > 0$,

$$P(|F - M_F| \geq r) \leq 4e^{-r^2/4},$$

where $M_F$ is the median of $F$.

**Theorem 1.** Any local aggregative game on a connected digraph $G$, where $G \in L(\Delta, n)$ and $\max\limits_{i} |A_i| \leq m$, admits a $10\Delta\sqrt{\ln(8mn)}$-PSNE.

*Proof.* The main idea behind the proof is to sample a random strategy profile from a mixed strategy Nash equilibrium of the game (any finite game is guaranteed to have a mixed strategy Nash equilibrium). Specifically, using a concentration argument, we show a *self-purification* [24; 23; 9] result that a profile sampled from a mixed strategy equilibrium is likely to be an approximate pure strategy equilibrium when the Lipschitz constant is small. This would typically be the case in large population aggregative games, where each player will likely have a sufficiently large number of neighbors. We follow the probabilistic approach outlined in [23], who introduced Lipschitz games and proved a remarkable result for general games with a small Lipschitz constant. We exploit the convexity of the payoff functions to obtain a tighter bound in the number of players $n$.

The proof involves three steps. First, using an application of the Talagrand's inequality, we will bound the deviation of each individual payoff function around their median value with high probability. Then, we will obtain a deviation around the mean by bounding the discrepancy between the median and the mean. Finally, we will take a union bound over all players to obtain an approximate pure strategy equilibrium by sampling from the mixed strategy equilibrium.

Let $\mu = (\mu_1, \ldots, \mu_n)$ be a mixed strategy Nash equilibrium of $G$, where each $\mu_i$ is a probability distribution on $A_i$. Then, the support of each $\mu_i$ contains all those pure strategies in $A_i$ that maximize the expected payoff of player $i$ when the other players play according to $\mu_{-i}$. Since the players play their mixed strategies independently, we obtain a product probability measure $P = \prod_i \mu_i$ over the strategy space $A = \prod_i A_i$.

Fix $\epsilon' > 0$ and let $\Delta = \epsilon'/5\sqrt{\ln(8mn)}$. We define event $E_{i,h,\epsilon'}$ to be the set of all strategy profiles $a$ such that by playing strategy $h$ against strategy profile $a_{-i}$ of her neighbors, player $i \in [n]$ receives nearly the same payoff as the median payoff $M_i(h)$ when it plays $h$ and others play their Nash equilibrium strategy:

$$E_{i,h,\epsilon'} = A_i \times \{a_{-i} \in A_{-i} : |u_i(h, f_G(a, i)) - M_i(h)| \leq \epsilon'\}.$$

Likewise we define event $E'_{i,h,\epsilon'}$ where we instead consider the mean payoff $\mathbb{E}_i(h)$ to player $i$ instead of the median payoff, when others play their Nash equilibrium strategy:

$$E'_{i,h,\epsilon'} = A_i \times \left\{ a_{-i} \in A_{-i} : \left| u_i(h, f_G(a, i)) - \underbrace{\int u_i(h, z)\mu_{-i}(dz)}_{\mathbb{E}_i(h)} \right| \leq \epsilon' \right\}.$$

We denote the complement of event $E$ by $E^c$. Since the payoff function $u_i(\cdot, \cdot)$ is convex and $\delta_i$-Lipschitz in the second argument (where $\delta_i \leq \Delta$) for every fixed instantiation of the first argument, we get from the Talagrand's inequality (following a scaling of $u_i$ by $\Delta$):

$$P(E_{i,h,\epsilon'}^c) \leq 4e^{-\epsilon'^2/4\Delta^2}. \tag{2}$$

Now, we can bound the deviation between the median $M_i(h)$ and the expected value $\mathbb{E}_i(h)$ using a standard result to obtain

$$P(E_{i,h,\epsilon'}^{'c}) \leq 8e^{-\epsilon'^2/16\Delta^2} < \frac{1}{mn},$$

for every $i \in [n]$ and $h \in A_i$. Since there are only $n$ players with at most $m$ strategies each, we immediately note on taking a union bound over players $i$ and their strategies $h \in A_i$, that there is some non-zero probability that the players play a mixed strategy $\epsilon'$-equilibrium. Therefore, we can sample a pure-strategy profile $a^*$ from the support of $\mu$ such that $a^* \in \cap_{i \in [n], h \in A_i} E_{i,h,\epsilon'}'$.

The arguments in [23] can be recycled to show that $a^*$ is a pure $2\epsilon'$-equilibrium. We reproduce these arguments for sake of completeness. Consider any player $i$ and strategy $h_i' \in A_i \setminus \{a_i^*\}$. Since $a^* \in E_{i,h_i',\epsilon'}'$ and the support of $\mu_i$ contains only those pure strategies in $A_i$ that maximize the expected payoff of player $i$ when others play according to $\mu_{-i}$, we note that

$$
\begin{aligned}
u_i(h_i', f_G(a^*, i)) &\leq \int u_i(h_i', z)\mu_{-i}(dz) + \epsilon' && \left(\text{since } a^* \in E_{i,h_i',\epsilon'}'\right) \\
&\leq \int u_i(a_i^*, z)\mu_{-i}(dz) + \epsilon' && (\text{since } a_i^* \text{ is in the support of } \mu_i) \\
&\leq u_i(a_i^*, f_G(a^*, i)) + 2\epsilon' && \left(\text{since } a^* \in E_{i,a_i^*,\epsilon'}'\right)
\end{aligned}
$$

The proof is complete since our choice of $i$ and $h_i'$ was arbitrary, and $\Delta = \epsilon'/5\sqrt{\ln(8mn)}$. $\square$

### 7.1.1 Extending the result to submodular functions

The result of Theorem 1 can be extended to submodular functions. Balcan and Harvey [27] proved the following result for a certain class of submodular functions. Specifically,

**Theorem (Balcan & Harvey [27]).** Let $F : 2^{[n]} \to \mathbb{R}^+$ be a non-negative, monotone, submodular, 1-Lipschitz function, and let $X \in [n]$ have a product distribution. Then for any $b, t \geq 0$,

$$P(F(X) \leq b - t\sqrt{b}) \cdot P(F(X) \geq b) \leq e^{-t^2/4}.$$

Defining $m = b - t\sqrt{b}$, and setting $b = 1$, this immediately yields the following for all $t \geq 0$:

$$P(F(X) \leq m)P(F(X) \geq m + t) \leq e^{-t^2/4}.$$

Let $M_F$ be the median of $F$. Since $P(F \geq M_F) = 1/2 = P(F \leq M_F)$, invoking this inequality twice with $m = M_F$ and $m = M_F - t$, where $t \geq 0$, we immediately get the following concentration inequality which is of identical form as the Talagrand concentration result in Lemma 7.1:

$$P(|F - M_F| \geq t) \leq 4e^{-t^2/4}.$$

Therefore, the result regarding existence of pure strategy Nash equilibrium under convex Lipchitz assumption on the individual payoff functions in Theorem 1 carries over nicely to non-negative, monotone, submodular, Lipschitz functions as well.

## 7.2 Information theoretic lower bounds

**Lemma 2.** Consider a directed random graph $DG(n, p)$ where $p \in (0, 1)$ is the probability of choosing any directed edge independently of others. Define $q = 1 - p$. Let $P_n$ be the probability that $DG$ is connected. Then, the probability that $DG$ is disconnected is $1 - P_N = nq^{2(n-1)} + O\left(n^2 q^{3n}\right)$.

*Proof.* Gilbert [25] proved an elegant result for bounding the probability that an Erdős-Rényi graph is disconnected. We will adapt his proof for our setting. Consider any node $i$. For $i$ to be in a component of size $k$, there must be exactly $(k - 1)$ other nodes in the component. Moreover, there should not be any edge between this component and any of the other $(n - k)$ nodes. The number of these missing edges is exactly $2k(n - k)$ when we account for the direction. Therefore, we must have the following recurrence relation:

$$1 - P_n = \sum_{k=1}^{n-1} \binom{n-1}{k-1} P_k q^{2k(n-k)} \tag{3}$$

First we bound this quantity from above. [25] noted that since $x(N - x)$ is convex, we have:

$$2k(n - k) \geq \begin{cases} (n - 2)k + n & \text{if } 1 \leq k \leq n/2 \\ (n - 2)(n - k) + n & \text{if } n/2 \leq k \leq n - 1. \end{cases} \tag{4}$$

Since $q < 1$, we can decompose the sum on right side of (3) into two sums to obtain

$$1 - P_n \leq \sum_{k=1}^{n-1} \binom{n-1}{k-1} q^{2k(n-k)} \leq \underbrace{\sum_{k=1}^{n/2} \binom{n-1}{k-1} q^{2k(n-k)}}_{(A)} + \underbrace{\sum_{k=n/2+1}^{n-1} \binom{n-1}{k-1} q^{2k(n-k)}}_{(B)}.$$

We will bound these two sums $(A)$ and $(B)$ separately. Note that, using (4),

$$
\begin{aligned}
(A) &= \sum_{k=1}^{n/2} \binom{n-1}{k-1} q^{2k(n-k)} \leq \sum_{k=1}^{n/2} \binom{n-1}{k-1} q^{(n-2)k+n} = \sum_{j=0}^{n/2-1} \binom{n-1}{j} q^{(n-2)(j+1)+n} \\
&= q^{2(n-1)} \sum_{j=0}^{n/2-1} \binom{n-1}{j} q^{(n-2)j} \leq q^{2(n-1)} \left[ \sum_{j=0}^{n-1} \binom{n-1}{j} q^{(n-2)j} - \binom{n-1}{n-1} q^{(n-2)(n-1)} \right] \\
&= q^{2(n-1)} \left[ \left(1 + q^{(n-2)}\right)^{n-1} - q^{(n-2)(n-1)} \right].
\end{aligned}
$$

Moreover, since $\binom{n-1}{k-1} = \binom{n-1}{n-k}$, using (4),

$$
\begin{aligned}
(B) &= \sum_{k=n/2+1}^{n-1} \binom{n-1}{k-1} q^{2k(n-k)} \leq \sum_{k=n/2+1}^{n-1} \binom{n-1}{n-k} q^{(n-2)(n-k)+n} \\
&= q^n \sum_{k=n/2+1}^{n-1} \binom{n-1}{n-k} q^{(n-2)(n-k)} = q^n \sum_{j=1}^{n/2-1} \binom{n-1}{j} q^{(n-2)j} \\
&= q^n \left[ \sum_{j=0}^{n/2-1} \binom{n-1}{j} q^{(n-2)j} - 1 \right] \leq q^n \left[ \left(1 + q^{(n-2)}\right)^{n-1} - 1 \right].
\end{aligned}
$$

Adding (A) and (B), we get an upper bound on the probability that the graph is disconnected:

$$1 - P_n \leq q^{2(n-1)} \left[ \left( 1 + q^{(n-2)} \right)^{n-1} - q^{(n-2)(n-1)} \right] + q^n \left[ \left( 1 + q^{(n-2)} \right)^{n-1} - 1 \right]. \quad (5)$$

On the other hand, the lower bound denotes the probability that some node is *isolated*, i.e. it does not have any incoming or outgoing edge. In other words, the lower bound $L$ is dictated by the union of the individual events $E_i$ that the node $i$ is isolated. Feller attributes the following inequality to Bonferroni [26]:

$$L \geq \sum_i P(E_i) - \sum_{i<j} P(E_i E_j).$$

Clearly, $P(E_i) = q^{2(n-1)}$ since $i$ does not share any edges in either direction with the remaining $(n-1)$ vertices. Now, the event $E_i \cap E_j$ happens when nodes $i$ and $j$ do not have any edge between them, and with any of the other $(n-2)$ vertices. Therefore, the total number of missing edges is $2 + 2 * 2(n-2) = 2(2n-3)$. Since there are $\binom{n}{2}$ such pairs $(i, j)$, we immediately get the lower bound:

$$1 - P_n \geq L \geq n q^{2(n-1)} - \binom{n}{2} q^{2(2n-3)} = n q^{2(n-1)} \left[ 1 - \left( \frac{n-1}{2} \right) q^{2(n-2)} \right]. \quad (6)$$

The statement of the lemma follows by combining the bounds from (5) and (6). $\qquad \square$

**Theorem 3.** Let $\epsilon > 0$, and $\alpha, \delta \in (0, 1)$. Let $n$ be the number of players in a local aggregative game, where each player $i \in [n]$ is provided with some convex $\Delta$-Lipschitz function $u_i$ and an aggregator $A$. Let $D_n \triangleq D_n(\Delta, \epsilon, A, (u_i)_{i\in[n]})$ be the sufficient in-degree (number of incoming edges) of each player such that the game admits some $\epsilon$-PSNE when the players play to maximize their individual payoffs $u_i$ according to the local information provided by the aggregator $A$. Assume any non-negative monotone submodular cost function on the edge set cardinality. Then for any $d \geq \max\{D_n, n^\alpha \ln n\}/(1-\alpha)$, any randomized algorithm that approximates the game structure to a factor $n^{1-\alpha}/(1+\delta)d$ requires exponentially many queries under the value oracle model.

*Proof.* The main idea is to construct a directed graph that has exponentially many spanning directed subgraphs, and define two carefully designed submodular cost functions over the edges of the graph, one of which is deterministic in query size while the other depends on a distribution. We will make it hard for a deterministic algorithm to tell one cost function from the other. This general paradigm [29; 30; 31] can be accomplished by ensuring two conditions: (a) these cost functions map to the same value on *almost* all the queries, and (b) the discrepancy in the optimum value of the functions (on the optimum query) is massive. Thus, since the total number of queries is exponential, we would make it difficult for the non-optimal function to figure out the optimal query when the optimal query would be chosen from a distribution over a large collection of the spanning subgraphs that satisfy the degree constraint with high probability (and thus, in turn, guarantee a pure strategy $\epsilon$-Nash equilibrium). Our analysis falls under the general framework introduced in [14], where lower bounds on some well known combinatorial problems were proved.

We first construct an good graph instance for our setting. Fix $\alpha$. Specifically, we consider $n^\alpha$ cliques $DG_1, \ldots, DG_{n^\alpha}$, each with $n^{1-\alpha}$ vertices. We form a spanning graph DG(V, E) by choosing an arbitrary vertex from each clique and then joining these vertices together via edges of arbitrary orientation. We now construct a random subset of edges $DR = \bigcup_{i \in [n]} DR_i$, where each $DR_i$ is obtained by randomly sampling every edge in $DG_i$ independently with probability $p = d/n^{1-\alpha}$. Since each $DG_i$ is a clique on $n'_i \triangleq n^{1-\alpha}$ vertices, and each edge is sampled independently with probability $p$, we can invoke Lemma 2 on each subset $DG_i$ separately. Then, taking a union bound, it is easy to see that the probability that $DR$ is connected is at least $1 - n \exp\left(-\Omega(n^\alpha \ln n (1 - o(1)))\right)$.

We now claim that with high probability the degree of each vertex in $DG_i$ restricted to edges in $DR_i$ is at least $D_n$. Let $deg_i(v)$ be the in-degree of any node $v$ restricted to set $DR_i$. Invoking Chernoff's bound on each $DG_i$, we have

$$P(\exists v \in DG_i : |deg_i(v) - n'_i p| \geq \alpha n'_i p) \leq \underbrace{2n' \exp(-\alpha^2 n'_i p/3)}_{\delta_1}.$$

Equivalently, with probability at least 1-$\delta_1$, we have for all $v \in DG_i$:

$$deg_i(v) \in [(1-\alpha)n_i'p, (1+\alpha)n_i'p] = [(1-\alpha)d, (1+\alpha)d].$$

Since $d \geq \dfrac{D_n}{1-\alpha}$, this immediately implies that with high probability, $deg_i(v) \geq D_n$, $\forall v$ in $DG_i$. Therefore taking a union bound over $DG_i, \ldots, DG_{n^\alpha}$, with probability at least $1 - n^\alpha \delta_1$, or equivalently $1 - 2n\exp(-\alpha^2 d/3)$, we have that degree of each vertex restricted to $DR$ is at least $D_n$. Since $d = \Omega(n^\alpha \ln n)$, this bound holds with high probability. This would ensure the existence of an $\epsilon$-Nash equilibrium in the underlying local aggregative game.

Thus far, we have shown a high probability result that DG(V, E), when restricted to $DR$, simultaneously satisfies both the spanning and degree constraints. Fix $\delta$. We denote the complement of a subset $S \subseteq E$ by $S^c = E \setminus S$. We now define two submodular functions $f_{DR}, g : 2^E \to \mathbb{R}^+$ that score any query $Q \in 2^E$. The cost of optimal solution in $f_{DR}$ is

$$\begin{aligned}
f_{DR}(Q) &= \min\left\{|Q \cap DR^c| + \min\{|Q \cap DR|, (1+\delta)npd\}, nd\right\} \\
g(Q) &= \min\{|Q|, nd\}.
\end{aligned}$$

Since $|Q| = |Q \cap DR| + |Q \cap DR^c|$, we have $f_{DR}(Q) \leq g(Q)$ for all $Q$. Moreover, since with high probability $DR$ is connected, the cost of optimal spanning graph under $f_{DR}$ is $(1+\delta)npd$. On the other hand, the optimal cost under $g$ is $nd$. Therefore, we have with high probability that the ratio of the optimal cost in $g$ and that in $f_{DR}$ is at least $\dfrac{1}{(1+\delta)p} = \dfrac{n^{1-\alpha}}{(1+\delta)d}$.

Note that since $f_{DR}(Q) \leq g(Q)$ for all $Q$, we have $P(f_{DR}(Q) \neq g(Q)) = P(f_{DR}(Q) < g(Q))$. Now we claim that the size of optimal query $Q^*$ is $nd$. To see this, consider first the case $|Q| \geq nd$. We have $g(Q) = nd$. Therefore,

$$P(f_{DR}(Q) < g(Q)) = P(\min\left\{|Q \cap DR^c| + \min\{|Q \cap DR|, (1+\delta)npd\}, nd\right\} < nd).$$

Clearly, this probability increases when we reduce the size of $Q$. Since $|Q| \geq nd$, we must have $Q^* = nd$. Now consider the other side, i.e. $|Q| \leq nd$. In this case, $g(Q) = |Q|$ and

$$f_{DR}(Q) = |Q \cap DR^c| + \min\{|Q \cap DR|, (1+\delta)npd\}.$$

Therefore, since we sampled the edges randomly, we see via an application of Chernoff's bound that

$$\begin{aligned}
P(f_{DR}(Q) < g(Q)) &= P(|Q \cap DR^c| + \min\{|Q \cap DR|, (1+\delta)npd\} < |Q|) \\
&= P(\min\{|Q \cap DR|, (1+\delta)npd\} < |Q \cap DR|) \\
&= P((1+\delta)npd < |Q \cap DR|),
\end{aligned}$$

increases when $|Q \cap DR|$ increases which happens when $|Q|$ increases. Therefore, we must have $|Q^*| = nd$ in this case as well. Also,

$$\begin{aligned}
P(f_{DR}(Q) < g(Q)) &= P((1+\delta)\mathbb{E}|Q \cap DR| < |Q \cap DR|) \\
&\leq \exp(-\delta^2 npd/3),
\end{aligned}$$

which is exponentially small. In other words, the probability that $f_{DR}$ and $g$ can be distinguished by an arbitrary query is exponentially small. The result stated in the theorem then follows immediately from the Yao's minimax principle. $\square$

## 7.3 Stability in $\gamma$-aggregative games

**Theorem 4.** Let $\gamma \in (0, 1)$, and $g_G(\cdot, \cdot, \ell, \cdot)$ be the $\gamma$-aggregator defined above. Let $P_G(\ell)$ be the property *"the number of maximum permissible intermediaries in a shortest path of length l in G"*. Then, $g_G$ is a $(2\eta\kappa_G, \eta\kappa_G)_{P,W_G,L}$- SSA, where $L < W_G$ and $\kappa_G$ depends on $\gamma$ and $W_G - L$.

*Proof.* The proof proceeds in three steps. First step is to show the existence of an approximate pure strategy Nash equilibrium under $P_G(D_G)$. The proof from Theorem 1 carries over directly while noting that we now instead need to use $\delta^\gamma(G)$ as the Lipschitz constant. The second step is to show

that the players are in an approximate pure strategy equilibrium when the aggregator now aggregates using $P_G(L)$. To see this, note that since under $P_G(W_G)$ we have an approximate pure $\epsilon'$-equilibrium profile $a^*$, and so for any strategy $h_i' \in A_i$ (note that Theorem 1 proved the following result for any deviation from $a_i^*$, however, the result holds trivially for $h_i' = a_i^*$ as well, and so we can consider an arbitrary action in $A_i$):

$$u_i(h_i', g_G(a^*, \gamma, W_G, i)) \le u_i(a_i^*, g_G(a^*, \gamma, W_G, i)) + \epsilon'. \tag{7}$$

Define $B(\gamma, \ell) \triangleq \sum_{t=0}^{\ell} \gamma^t$, and $C(a, \gamma, \ell, i) = \sum_{t=0}^{\ell} \gamma^t f_G^t(a, i)$. Then, since the payoff functions $u_i$ are $\eta$-Lipschitz in the second argument, it is easy to show that for every $a_i'$,

$$|u_i(a_i', g_G(a^*, \gamma, \ell, i)) - u_i(a_i', g_G(a^*, \gamma, W_G, i))| \le \eta \kappa_G(\gamma, \ell),$$

where

$$\kappa_G(\gamma, \ell) \triangleq \max_i \left| C(a^*, \gamma, \ell, i) \left( \frac{B(\gamma, W_G) - B(\gamma, \ell)}{B(\gamma, W_G)B(\gamma, \ell)} \right) - \frac{C(a^*, \gamma, W_G, i) - C(a^*, \gamma, \ell, i)}{B(\gamma, W_G)} \right|.$$

In particular, substituting $a_i' = h_i'$, we have

$$u_i(h_i', g_G(a^*, \gamma, L, i)) \le u_i(h_i', g_G(a^*, \gamma, W_G, i)) + \eta \kappa_G(\gamma, L). \tag{8}$$

Also, substituting $a_i' = a_i^*$ and using the other direction, we have

$$u_i(a_i^*, g_G(a^*, \gamma, W_G, i)) \le u_i(a_i^*, g_G(a^*, \gamma, L, i)) + \eta \kappa_G(\gamma, L). \tag{9}$$

Combining (7), (8) and (9), we get that players are playing an $(\epsilon' + 2\eta \kappa_G(\gamma, L))$-PSNE under $P_G(L)$ by sticking to the profile $a^*$:

$$
\begin{aligned}
u_i(h_i', g_G(a^*, \gamma, L, i)) &\le u_i(h_i', g_G(a^*, \gamma, W_G, i)) + \eta \kappa_G(\gamma, L) \\
&\le u_i(a_i^*, g_G(a^*, \gamma, W_G, i)) + \epsilon' + \eta \kappa_G(\gamma, L) \\
&\le u_i(a_i^*, g_G(a^*, \gamma, L, i)) + \epsilon' + 2\eta \kappa_G(\gamma, L).
\end{aligned}
\tag{10}
$$

Finally, exploiting the $\eta$-Lipschitz property again, we immediately get that the payoff of each player does not decrease too much under $P_G(L)$:

$$u_i(a_i^*, g_G(a^*, \gamma, L, i)) \ge u_i(a_i^*, g_G(a^*, \gamma, W_G, i)) - \eta \kappa_G(\gamma, L).$$

Therefore, $g_G$ is $(2\eta \kappa_G(\gamma, \ell), \eta \kappa_G(\gamma, \ell))_{P, W_G, L} -$ structurally stable. $\qquad \square$

Note that $\kappa_G(\gamma, \ell)$ is often small when $W_G - L$ is small. For instance, in large population games with the average aggregator $f_G^t(\cdot, \cdot)$, for a fixed $\gamma$, $C(\gamma, \ell, i)$ would typically be small compared to $B(\gamma, \ell)$ since the $f_G^t(a, i)$ would generally be much smaller than 1. Then, $\kappa_G$ is largely determined by the gap between $B(\gamma, \ell)$ and $B(\gamma, W_G)$ which would be small when $\ell$ is close to $W_G$. In particular, $\kappa_G$ would be small when either $\gamma$ is small (in which case the aggregator behaves increasingly like a local aggregator), or when $\ell$ is close to $W_G$. Thus, by controlling $\ell$ and $\gamma$, we can ensure $\kappa_G$ is small.