[Reviews · NeurIPS 2017]

Reviewer 1



This work extends learning for contextual potential games beyond undirected tree structures to local aggregative games in which payoffs are based on the aggregate strategy of a player’s neighbors and the player’s own strategy. The paper proves that approximate pure strategy Nash equilibria exist for all games with limits on their Lipschitz constants. A hardness result for the number of queries needed to learn the game structure from observed Nash equilibria is developed in Theorem 3. Has progress been made on the elusive proof eluded to in Line 41? As described in Page 4, this seems like a critical missing component of the submitted work. Despite the seemingly unresolved negative results of Theorem 3, a margin-based approach is employed to recover the structure and payoffs of the game. The real experiments on political “games” are interesting and show reasonable results. However, the “ground truth” for these settings is open to debate (e.g., Breyer is often considered more conservative than Sotomayor and a more natural connector from Liberals to Conservatives). The experimental results could be improved by generating synthetic games and showing that the true structure is recovered. My assessment for this paper is mixed. Though the method seems to work well in practice, the underlying theory has not been fully developed in a way to resolve the hardness result of Theorem 3. ---- Thank you for your response. I look forward to synthetic experiments and also suggest compressing Figure 4 and 5 so that more of the description for this datsaset can be placed into the main text.

Reviewer 2



Summary: The authors consider games among n players over a digraph G=(V,E), where each player corresponds to a node in V, the payoff function of each player depends on the choices of its own and those of neighbors. A further assumption is that the payoff function is Lipschitz and convex (or submodular). For games with sufficiently smooth payoff functions, the authors show that an approximate PSNE exists. Also, the authors prove that there exists a digraph for which any randomized algorithms approximating PSNE needs exponentially many value-queries for a submodular payoff function. Then the authors consider a variant of the game called gamma-local aggregative game, for which the payoff function depends on not only neighbors but all other players under the condition that other players choices affect the payoff in the magnitude of gamma^(distance from the player). For such games, the authors show a sufficient condition that an approximate PSNE exists. They further consider a learning problem for the games where, given approximate PSNEs as examples, the learner’s goal is to predict the underlying digraph. An optimization problem for the learning problem is proposed. Finally, the authors show experimental results over real data sets. Comments: The proposed game is a generalization of the aggregative game to a graph setting. The technical contribution of the paper is to derive sufficient/necessary conditions for the existence of approximate PSNEs for the games. On the other hand, the theoretical results for learning graphs from approximate PSNEs is yet to be investigated in the sense that no sample complexity is shown. However, the proposed formulation provides interesting results over real data sets. I read the authors' rebuttal comments and I would appreciate if the authors reflect your comments to me in the final version.

Reviewer 3



This paper presents theoretical analyses of local aggregative games that covers a wide range of multi-agent interaction scenarios. It is an important problem. The analyses are provided from multiple aspects, such as the existence of epsilon-Nash equilibrium, information theoretical lower bound, and the stability of its generalized gamma-aggregative games. An algorithm is also provided on learning the digraph structure and payoffs from data. The analyses are novel and seem to be correct (although having not checking in details). My issues mostly lie on the empirical analysis section, where although some experiments are carried out, it is still unclear how the proposed learning algorithm perform both quantitatively and and qualitatively, and how these experiments are connected to the aforementioned analyses.